# CARs—A New Perspective to HCMV Treatment

**DOI:** 10.3390/v13081563

**Published:** 2021-08-07

**Authors:** Christopher Bednar, Armin Ensser

**Affiliations:** Institute of Clinical and Molecular Virology, University Hospital Erlangen, Friedrich-Alexander Universität Erlangen-Nürnberg, 91054 Erlangen, Germany; Christopher.Bednar@uk-erlangen.de

**Keywords:** human cytomegalovirus (HCMV), antiviral immunotherapy, chimeric antigen receptor (CAR), cytotoxic T-lymphocytes, adoptive T-cell therapy

## Abstract

Human cytomegalovirus (HCMV), by primary infection or reactivation, represents a great risk for immune-suppressed or compromised patients. In immunocompetent humans, the immune system suppresses the spread of HCMV during an infection, resulting in a mostly asymptomatic or mild course of the disease, whereas in immune suppressed patients, the compromised host immune response cannot control the viral infection. Multiple viral immunomodulatory mechanisms additionally contribute to immune evasion. Use of chimeric antigen receptors (CARs), a treatment strategy adapted from cancer immunotherapy, is investigated for possible application to combat HCMV and other infections in immunocompromised patients. The administration of CAR^+^ T-cells directed against HCMV antigens can bypass viral immune evasion and may complement existing treatment methods. This review gives a short overview of HCMV, the obstacles of current treatment options as well as a brief introduction to CARs and the current research situation on CAR^+^ T-cells against HCMV.

## 1. Evasion of the Host Immune System by HCMV

Human cytomegalovirus (HCMV) is a highly prevalent pathogen, infecting the majority of the adult population followed by lifelong persistence in the host. In most individuals, the human immune system is able to overcome the infection and suppress reactivation from latency without emergence of clinical symptoms. For immunosuppressed patients, i.e., during solid organ or allogeneic stem cell transplantation, HCMV infection poses a serious threat and can cause major clinical complications, and induce graft rejection and enhance graft versus host disease (GvHD), respectively. Symptoms ranging from the inflammation of several organs to systemic disease result in an increased mortality rate [1,2]. Antiviral substances, in particular (Val) Ganciclovir (GCV) and, recently, Letermovir, have improved the clinical situation, but are limited by the emergence of drug resistant strains, as well as, for GCV, bone marrow suppression and nephrotoxicity [3,4]. Successful control of CMV infection or reactivation requires establishment of specific adaptive immune responses in the patient. In addition to necessary therapeutic immune suppression, this is counteracted by the ability of HCMV to deter the host immune system by various mechanisms. Cytotoxic T-cells are no longer able to recognize HCMV-infected cells via peptide-presentation by the major histocompatibility complex (MHC) due to downregulation of human leukocyte antigen (HLA) expression by HCMV [5,6]. Viral Fc receptors (FcRs), three in HCMV, and more than one in MCMV, are expressed on the surface of infected cells, which compromise antibody-dependent cellular cytotoxicity (ADCC) by the binding of the Fc domain of host antibodies to viral FcRs [7]. An accepted but complex treatment option is the administration of ex vivo expanded HCMV-specific cytotoxic T-cells from HCMV-positive donors. This approach has shown promising results [8,9], but the clinical application is limited due to requirement for HLA matching of HCMV-specific cytotoxic T-cells of the donor to the recipient, as well as the labor-intensive and time-consuming effort of expanding antigen-specific T-cells. Low precursor frequencies practically exclude this option in HCMV-naïve settings (HCMV seronegative stem cell donors or HCMV seronegative recipients of HCMV-positive organs). An HCMV vaccine would be advantageous in those settings to immunize donors or recipients in those high-risk constellations, respectively, but has not been approved. However, recent studies have shown that administration of CMV-specific T-cells (CMVSTs) from a carefully selected pool of third-party donors with an HLA match of only 2 out of 8 can result in a strong antiviral response [10]. These findings allow cryopreserved ‘cell banks’ to be manufactured, at least in the case of common HLA profiles, resulting in an off-the-shelf treatment option for serious CMV-infections. Still, seropositive donors with an at least partially matching HLA-type are required to create CMVSTs, whereas T-cells carrying a CMV-specific chimeric antigen receptor (CAR) can be manufactured from any donor, including HCMV negative donors in stem cell transplantation. CARs that retarget the specific cytolytic potential of cytotoxic T-cells, and have meanwhile proven their efficacy in cancer immunotherapy, are the focus of this review.

## 2. History of CARs

The antigen-specific activation of cytotoxic T-cells via binding to the T-cell receptor (TCR) was a model for the creation of CARs which are able to induce T-cell cytotoxicity through binding of a designated antigen in an HLA-independent manner [11]. The first CARs were fusions proteins of an antigen binding domain, derived from the Fv domain of an immunoglobulin (=single chain variable fragment, scFv) directed against a specific target antigen which was linked to the intracellular signaling domain CD3ζ from the T-cell receptor (first generation). Upon binding of the scFv to the specific antigen, the CAR triggers a signal transduction cascade in the CAR^+^ T-cell resulting in T-cell activation, cytokine secretion and lysis of the antigen presenting target cell [12]. One significant advantage over conventional T-cell therapy is the independence of T-cell activation from antigen-presentation by HLA, circumventing the need of HLA haplotype matching of the patient. First generation CARs were limited by the amount of required cytokine production for cell proliferation as well as the lack of proliferation of resting T-cells [13]. The introduction of a co-stimulatory domain (CD28, 4-1BB, etc.) showed that CARs containing at least one co-stimulatory domain in addition to the activating intracellular CD3ζ domain persist significantly longer due to proliferation triggered by the activated co-stimulatory domain [14,15]. These second-generation CARs are currently and most commonly used in CAR research [16]. Clinical studies using CARs targeting CD19^+^ B-cell lymphoma have led to the approval of tisagenlecleucel as an immunotherapy product in several countries [17,18,19,20]. Third-generation CARs were created by combination of several co-stimulatory domains, and have been tested in few clinical studies. Although first results in patients look promising further studies of this CAR design are required [21,22]. When targeting solid tumors, CAR^+^ T-cells face evasion by the loss of the target antigen on tumor cells, making them invisible to CAR^+^ T-cells preventing target cell lysis. Therefore, CAR^+^ T-cells have been further modified so that they express transgenic products, such as pro-inflammatory cytokines upon activation, which accumulate in the targeted tissue. This can recruits a second wave of immune cells to the respective tissue that may also target tumor cells that have become invisible to CAR^+^ T-cells [23]. Such constructs are designated T-cells redirected for universal cytokine killing (TRUCKs) and make up the fourth generation of CARs.

A novel design, termed *convertible*CARs aims to establish a universal CAR platform by replacing the antigen-specific scFv for an inert form of the ligand-binding domain of a natural killer cell-activating receptor (iNKG2D). Activation of *convertible*CAR™ T-cells only occurs when combined with a bispecific adapter comprised of the ligand of iNKG2D fused to a specific antigen-targeting antibody. This system requires only one type of CAR^+^ T-cell, which can be adapted to target any desired antigen by combing the administration of *convertible*CAR™ T-cells with the respective modified antibody, circumventing the labor intensive process of designing and testing new CAR constructs restricted to only one target antigen [24]. A comparable but more direct approach is the usage of bispecific T-cell enhancers (BiTEs), which are hybrid molecules consisting of two scFvs coupled by a linker peptide. One scFv is directed against the antigen of choice whereas the other binding domain binds specifically to the CD3 molecule on the surface of T-cells. Upon binding to the target on the antigen presenting cell, as well as the T-cell receptor on the cytotoxic T-lymphocyte, the T-cells are activated and exert effector functions. These BiTEs are administered as a soluble protein, which obviates the requirement of genetic generation and expansion of CAR^+^ T-cells. From the here presented options in different designs (Figure 1), second-generation CARs are the current construct of choice in research of CARs targeting HCMV.

## 3. Viral Glycoproteins as Markers of Infected Cells

One major obstacle in the development of novel CAR constructs in cancer therapy is the requirement for a strictly tumor-specific target antigen to ensure high on-target effects while retaining low off-target cytotoxicity. In most cases, tumor-specific antigens are also expressed on a variety of other human cell types, making also healthy cells targets of CAR^+^ T-cells and resulting in unwanted tissue damage. This disadvantage can be turned into an advantage by the introduction of CARs into antiviral therapy. Enveloped viruses generally encode specific surface proteins, enabling them to bind and enter host cells. During replication, the same and additional viral surface proteins are presented on the infected cell surface, at least transiently. HCMV-infected cells present a particular variety of viral glycoprotein complexes on their surface, which may serve as potential targets for CAR-immunotherapy. The viral surface is clustered with various heterogeneous glycoprotein complexes (GC), including of the oligomer gB (GC I), the heterodimer gN/gM (GC II), the heterotrimer gH/gL/gO (GC III) as well as the so-called pentameric complex (PC) comprised of gH/gL/UL128/UL130/UL131a (Figure 2). Although not every function of these different complexes is completely understood, it was shown that these envelope proteins play an essential role in attachment and entry into numerous cell types as well as in the viral life cycle (reviewed in [25]). Multiple neutralizing and non-neutralizing antibodies targeting the different glycoprotein complexes have been isolated and characterized from HCMV-positive individuals, providing the basis for the design of novel CAR constructs [26,27]. The glycoprotein gB is expressed during the early or delayed early phase of HCMV replication, even before one replication cycle is completed. Proteolytic cleavage of gB by a cellular protease at the cell surface is required for maturation, precluding viral evasion by complete gB downmodulation. This makes gB an attractive target for CAR-immunotherapy since infected cells could be detected and lysed even before new HCMV virions are released to infect more cells [28]. Because herpesviral protein expression is extremely downregulated in viral latency, the glycoprotein complexes are only presented on the cell surface in lytic and productive states of the infection [29]. Concluding from these observations, eradication of HCMV from infected hosts will never be achieved by CAR^+^ T-cells targeting the viral glycoprotein complexes due to the lack of their expression during latent stages. Nevertheless, since CAR^+^ T-cell therapy is supposed to be applied during severe acute lytic HCMV-infection or re-activation in immunocompromised persons, this critical lytic phase would be suppressed by CAR^+^ T-cells, thus rescuing the patient from severe sequelae or even death.

## 4. HCMV-Specific CARs

The first successful approach to create an HCMV-specific CAR was conducted by Full et al. [30]. The generated CAR (27-287) consisted of a scFv derived from a monoclonal antibody directed against the glycoprotein gB of HCMV, linked by an immunoglobulin-hinge domain to the intracellular costimulatory domain of CD28 and the activating domain of CD3ζ. In that study, the surface expression of gB could be confirmed via fluorescence-activated cell sorting (FACS) 48 h after HCMV infection of primary human foreskin fibroblast (HFF) cells, which are an established model for HCMV-infection studies. This confirmed that HCMV-gB is accessible on the surface of infected cells before the completion of one viral replication cycle (3–4 days) and before release of virus occurs. This observation made HCMV-gB a compelling target for CAR^+^ T-cell therapy, since infected cells can potentially be lysed by cytotoxic T-cells before the virus is able to infect more cells. CAR transfer by lentiviral transduction or mRNA electroporation of human T-cells resulted in a high surface expression of 27-287. Co-incubation of 27-287^+^ T-cells together with HCMV-gB-transfected 293T-cells resulted in cytotoxic lysis of these HCMV-gB-presenting cells. Co-incubation of 27-287^+^ T-cells together with HCMV-infected HFFs led to cytokine secretion (TNF and IFN-γ). It was assumed that CAR-immunotherapy targeting gB has the potential to limit the spread of HCMV in vivo [30].

However, it was observed that HCMV-infected cells can resist cytotoxic lysis by CAR^+^ T-cells presumably via viral effector proteins, which was investigated in more detail in studies by Proff et al. [31] Despite the strong surface expression of gB on infected cells, co-incubation of 27-287^+^ CAR T-cells together with HCMV-infected HFF cells resulted only in a low cytolytic activity. Comparative experiments with antigen-independent activation of CAR^+^ T-cells showed that ineffective lysis does not result from the low efficacy of T-cell activation, since cytokine secretion and degranulation were not impaired, but rather from a peculiarity of HCMV infection. Contrary to scientific consensus that the impaired lytic activity of activated T-cells is a result of reduced antigen presentation due to HLA-downregulation, experiments with CAR-independent activation of CAR^+^ T-cells against HFFs infected with a mutant HCMV-strain imply that HCMV has additional viral effector proteins, in particular UL37x1 and UL36, that impair T-cell cytotoxicity [31].

This was consistent with an earlier study which had observed that human CD4(+) T-cell clones specific for the major immediate-early protein IE1 exerted efficient perforin-based cytotoxicity against HCMV IE peptide-pulsed targets, but not when the peptide-pulsed targets were HCMV infected [32]. An elegant study combined recombinant MCMV-OVA and TCR-transgenic, SIINFEKL-Kb-specific OT-I cells to observe the in vivo killing capacity of cytotoxic T cells by 2-photon microscopy [33]. Modest killing of MCMV infected cells was only observed when MCMV was severely compromised by deleting 2 viral inhibitors (m06, m152) of MHC-I mediated antigen presentation, and was then similar to modified vaccinia virus Ankara expressing the same OVA antigen (MVA-OVA), a poxvirus which lacks 18 kb compared to vaccinia virus. It required Perforin and multiple dynamic CTL contacts per MCMV infected cell before target cells were killed. Despite earlier solid evidence that adoptive transfer of lymphokine activated killer cells, or as few as 20,000 MCMV experienced CD8^+^ CTLs, can prevent MCMV disease in irradiated mice [34,35], killing of target cells infected by an immune evasion competent MCMV-OVA was not seen in the model [33].

In their subsequent paper, Proff et al. showed that activated CAR^+^ T-cells inhibit HCMV replication by secretion of TNF and IFN-γ, underlining that interferons and cytokines may have a major role in controlling cytomegaloviruses [36]. One possibly weak spot in the structure of CARs used at that time was the CH2-CH3 Fc hinge domain, because it was shown that endogenous Fc receptors bind to the hinge domain, activating the CAR^+^ T-cell independently from target antigen recognition and therefore lowering their specificity [37]. By introducing specific mutations in the immunoglobulin-hinge domain, Proff et al. were able to turn this disadvantage into an advantage. HCMV encodes three own viral FcRs, which impair ADCC by binding antibodies at their Fc domain. These viral FcRs differ significantly in their mode of binding to Fc domains compared to endogenous FcRs, which makes them a potential specific target for immunotherapy. The generated mutated version of CAR 27-287 would specifically bind to viral FcRs, but not to endogenous cellular FcRs. In theory, this construct would be able to target infected cells both with its gB-binding scFv and with its viral FcR-specific lg-hinge domain. T-cells expressing the mutated CAR backbone construct with an unspecific scFv were indeed able to recognize HCMV-infected HFF cells. However, cytokine secretion was reduced compared to unmutated constructs, probably because of a generally reduced capacity for cytokine production in consequence of expression of a more unstable mutated Fc protein domain.

HCMV-gB was also chosen as the target antigen in a recent study by Olbrich et al. [38]. In this paper, the group designed CAR constructs based on the high-affinity gB-specific neutralizing antibody SM5-1. The CARs used in this study consisted of a scFv fused to linker sequences, connected to the different co-stimulatory domains CD28 or 4-1BB and ended in the activating domain CD3ζ. Cytotoxic T-cells were transduced with the constructs and the resulting CAR^+^ T-cells were tested in vitro and in vivo. As shown earlier by Full et al., co-incubation of gB-CAR^+^ T-cells together with transfected gB-expressing HEK293T cells also resulted in increased target cell death compared to control CAR^+^ T-cells. Cytokine secretion, degranulation, T-cell proliferation, target cell viability, and reduction of replication of a luciferase-expressing reporter virus (HCMV-GLuc) were then investigated after co-incubation of CAR^+^ T-cells together with HCMV-GLuc-infected mesenchymal stem cells (MSC). As expected, most gB-CAR^+^ T-cells showed increased IFN-γ secretion, proliferation, surface CD107a expression; the co-incubation with HCMV-infected MSCs resulted in increased killing of target cells as well as in a reduction of viral replication compared to control CAR^+^ T-cells. Notably, T-cells transduced with CAR constructs containing the 4-1BB co-stimulatory domain showed a weaker proliferation rate after activation compared to CD28-CARs, whereas in all other investigated categories, 4-1BB-CARs performed better than their CD28-containing counterparts. Because of the superior performance of 4-1BB-CARs, T-cells transduced with this construct were used for in vivo experiments in humanized mice. The animals used for these experiments were humanized by transplantation of CD34^+^ hematopoietic stem cells (HSCs) derived from human cord-blood several months before infection studies. Human MRC-5 fibroblasts were infected with HCMV-GLuc and administered to the mice via intraperitoneal injection. gB-CAR^+^ T-cells were administered intravenously eight weeks after infection of the mice. By using a luciferase reporter expressing HCMV, non-invasive observation of HCMV distribution in the animal by bioluminescence measurement could be achieved one and four weeks after administration of gB-CAR^+^ T-cells. Five out of eight tested mice treated with gB-CAR^+^ T-cells showed a significantly lower luminescence signal compared to untreated infected mice, and the anti-viral response correlated with the number of T-cells in the spleen.

Ali et al. [39] created eight different CAR constructs based on neutralizing antibodies targeting several HCMV-specific glycoproteins. These CARs consisted of a scFv directed against the PC, linked to the transmembrane domain of CD8 via IgG_4_ spacer sequence, leading into the co-stimulatory domain 4-1BB and ending in the activating domain CD3ζ. Primary human CD8^+^ cytotoxic T-cells were transduced and the effectiveness of the resulting CAR^+^ T-cells was investigated by measurement of cytokine secretion, proliferation and degranulation. Cytotoxicity and reduction of viral replication after co-incubation with ARPE-19 epithelial cells that were infected with a GFP-expressing reporter strain of HCMV. Two of the eight CAR constructs showed T-cell activation by increased secretion of IFN-γ and TNF-α as well as a high surface expression of CD107a. Only one of these two CARs, named 21E9, showed moderate T-cell proliferation and specific target cell killing after co-incubation with HCMV-infected ARPE-19 cells. Four different CARs showed reduction of viral replication to a certain extent, but only 21E9 mediated a constant antiviral activity across multiple experiments. However, no experiments were conducted to investigate whether the secreted cytokines act as primary effectors, as shown in previous studies, or if the antiviral properties are conferred by T-cell mediated cytotoxicity. Since ARPE-19 cells are not the cells of choice for an infection model of HCMV and may be less permissive to HCMV infection, it is certainly possible that HCMV does not express its whole arsenal of immune evasion genes, allowing easier target cell lysis for the CAR^+^ T-cells compared to the standard infection model of HFF cells.

## 5. Discussion of Other Antiviral CARs

The investigation of CARs as potential candidates for antiviral immunotherapy is not limited to HCMV. Research groups have also designed CARs, which target different surface antigens on cells infected with Human Immunodeficiency Virus 1 (HIV-1), Epstein-Barr–Virus (EBV), or Hepatitis B and C Virus (HBV and HCV).

Although combined anti-retroviral therapy (cART) currently remains the best treatment option for HIV-infected persons, the clearance of latent viruses remains a major problem to curing HIV-infection. To effectively apply the recently developed “kick and kill” strategies, which aim to activate latent viruses using latency reversing agents (LRAs) and kill infected cells by the host immune system, a potent immune response is necessary for the eradication of the HIV reservoir from the body. By incorporating the scFv of the broadly neutralizing HIV-1-specific antibody VRC01 into a CAR of the third generation, Liu et al. were able to create CAR^+^ T-cells targeting HIV-1-infected CD4^+^ T-cells [40]. In an in vitro infection model, re-emergence of viral infection after removal of cART was inhibited significantly by CAR^+^ T-cells and LRA-treated CD4^+^ T-cells isolated from infected patients were lysed efficiently by CAR^+^ T-cell mediated cytotoxicity.

Apart from HCMV, another clinically relevant member of the herpesviruses is EBV. EBV expresses an abundance of the viral glycoprotein gp350 on the surface of cells during lytic infection, which makes this glycoprotein a potential target for CAR^+^ T-cell therapy. A second generation CAR was created Slabik et al. based on a highly neutralizing antibody targeting gp350 [41]. The created CAR^+^ T-cells were cytotoxic against the EBV^+^ cell line B95-8 in vitro, specifically targeting gp350^+^ cells. In an in vivo infection model, the majority of infected mice controlled EBV spread and showed reduced inflammation as well as malignant lymphoproliferation when treated with gp350 CAR T-cells. However, most clinical problems with EBV are due to its transforming properties, causing lymphoproliferative disease and cancer, in which EBV-infected cells are in the latent phase and do not express gp350 nor other glycoproteins.

One major clinically relevant pathogen is HCV, infecting millions of people worldwide and causing severe chronic liver complications such as cirrhosis, liver failure, or hepatocellular carcinoma. Similar to HCMV, HCV exerts a plethora of evasion mechanisms to escape the host immune response, making the virus a candidate for antiviral CAR immunotherapy. Therefore, Sautto et al. designed a second generation CAR containing the scFv of the broadly neutralizing antibody e137, which targets HCV/E2, the major viral surface protein on HCV-infected cells [42] CAR^+^ T-cells were able to lyse the hepatocellular carcinoma cell line HuH-7.5 when infected with a cell culture strain of HCV and showed high levels of cytokine secretion, such as IFN-γ, TNF-α, and IL-2. The capabilities of the e137CAR T-cells in an in vivo infection model remain to be tested.

Another causative agent of hepatitis is HBV. Though antiviral substances can control an infection, they are not able to eradicate the virus from the host. HBV can persist in nuclei of infected cells in a covalently closed circular DNA (cccDNA) form, which is not targeted by antiviral nucleotide analogues. During acute hepatitis B, the patient exerts a strong T-cell immune response against viral surface antigens. This is not the case in chronically infected patients, but is necessary to clear the infection. This issue may be circumvented by CAR immunotherapy. Krebs et al. generated a CAR targeting the hepatitis B surface antigen (HBsAg), which is expressed on the surface of hepatocytes in chronic hepatitis B [43]. When co-incubated together with an HBV^+^ hepatoma cell line or HBV-infected primary human hepatocytes, HBsAg-specific CAR^+^ T-cells showed high levels of IFN-γ and IL-2 secretion, as well as specific lysis of infected cells. Most importantly, the level of HBV cccDNA was reduced by over 99.99%. In the subsequent paper, Krebs et al. aimed to test this CAR construct in a murine infection model [44]. Engrafted HBsAg-specific CAR^+^ T-cells were recruited to the site of infection and proliferated in HBV-infected mice. The expression of proinflammatory cytokines in livers of infected animals was strongly upregulated; the number of HBV^+^ hepatocytes as well as the viral load in the bloodstream was reduced drastically. It was shown that the engraftment of CAR^+^ T-cells resulted in only minor side effects, indicating that the benefits of this approach clearly outweigh the risk factors, paving the road towards an immunotherapy against chronic hepatitis B.

## 6. Conclusions

Only a few studies of CAR^+^ T-cells targeting cytomegaloviruses were conducted to date, and almost none in vivo. Considering the encouraging results from in vitro experiments, further investigation of the CAR approach as a treatment option in antiviral therapy should be intensified, in particular after the approval of tisagenlecleucel as a successful CAR-based approach against CD19^+^ B-cell lymphoma. The following table briefly summarizes aspects of CAR T-cell therapy in comparison to conventional T-cell therapy with regard to their potential as antiviral therapies (Table 1).

For pre-clinical in vivo studies, a suiting animal model is mandatory to investigate the on-target effects and off-target cytotoxicity of CAR^+^ T-cell treatment. One option would be the usage of humanized mice, as in the paper of Olbrich et al. [38], which is confined to co-transferred infected human target cells, an absence of systemic CMV spread, and cannot address the role of species-specific soluble and tissue factors. On the other hand, the murine pendant to HCMV, murine cytomegalovirus (MCMV) allows the combination of transgenic as well as knockout models; it is the accepted small animal model, but is nevertheless limited by significant genetic differences after more than 60 million years of divergent evolution of viruses, mice, and men. Nonhuman primate models could be established, but are costly, and would theoretically allow the direct evaluation of HCMV specific CARs if they are cross-reactive to the well conserved homologous Rhesus CMV glycoprotein B, such as the gB CAR designed by Full et al. [30]. In summary, more research in design, antiviral activity, and safety of CARs is needed to confirm the effectiveness of the CAR approach for application as an immunotherapy against HCMV; it has the potential to circumvent many known obstacles of conventional treatment and bypasses HLA-dependent viral immune evasion.

## Figures and Tables

**Figure 1 viruses-13-01563-f001:**
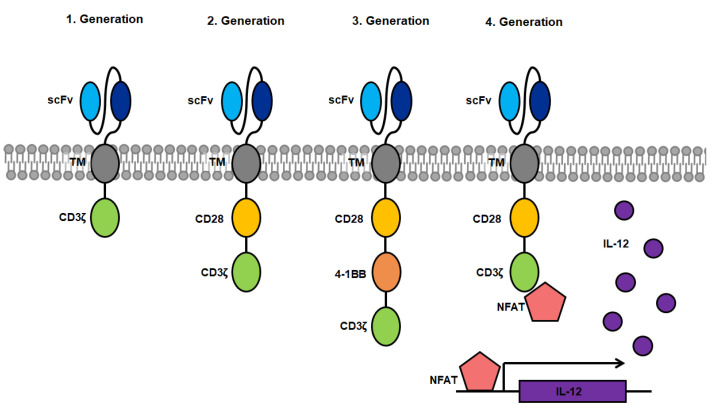
Overview of CAR designs from first to fourth generation. Blue: single-chain variable fragment (scFv) consisting of the heavy and light chain of the binding domain of antigen-specific antibodies coupled by a linker peptide; grey: transmembrane domain; yellow: co-stimulatory domain of CD28; orange: co-stimulatory domain of 4-1BB (CD137); green: activating domain consisting of the CD3ζ domain of the T-cell receptor; pink: nuclear factor of activated T-cells (NFAT); purple: interleukin-12 (IL-12) which is expressed after binding of NFAT to an NFAT-driven IL-12 cassette.

**Figure 2 viruses-13-01563-f002:**
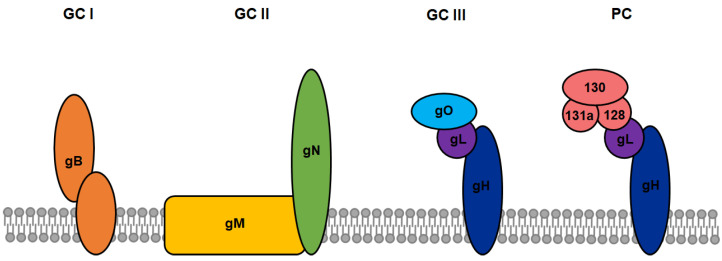
Various types of glycoprotein complexes on the surface of HCMV and the host cells. The viral surface is clustered with various heterogeneous CMV glycoprotein (g) complexes (GC), including of the oligomer gB (GC I), the heterodimer gN/gM (GC II), the heterotrimer gH/gL/gO (GC III), and the so-called pentameric complex (PC) comprised of gH/gL/UL128/UL130/UL131a.

**Table 1 viruses-13-01563-t001:** Comparison of antiviral CAR T-cells with conventional antigen-specific T-cell therapy.

	CAR T-Cell Therapy	Antigen Specific T-Cell Therapy
Antigen-specific target cell recognition	yes	yes
HLA-independent mode of action	yes	no
HLA-matching required	no	partial match
Seronegative donor possible	yes	no
Circumvention of CMV-mediated HLA-targeted immune evasion	yes	no
Isolation, ex vivo expansion and genetic modification of T-cells (mRNA or Lentiviral vector)	yes	no
Isolation and ex vivo expansion of specific T-cells required	no	yes
‘Off-the-shelf’ therapy possible	yes	HLA dependent
Eradication of virus from host	no	no

## Data Availability

Not applicable.

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
