# Peer review of "CARs—A New Perspective to HCMV Treatment"

_viruses, 2021, doi:10.3390/v13081563_

Round 1

Reviewer 1 Report

This review summarizes current chimeric antigen receptor (CAR)-T cells against human cytomegalovirus (HCMV). The authors give sufficient introductions of HCMV, CARs and viral glycoproteins on the infected cells. The most interesting part is the section 4: HCMV-specific CARs, which introduces the current studies in the area. Overall, the review is written in a clear and concise manner. 

Main concerns:

  1. HCMV latency and persistence. Depending on host physiology and the cell types infected, CMV persistence comprises latent, chronic, and productive states that may occur concurrently. CAR-T cell therapy of HCMV may vary during different stages, which is not well discussed.
  2. Advantages and disadvantages. The authors need to give a list or summary presenting the advantages and disadvantages on HCMV CAR-T cell therapy. Clearly, it will not eliminate HMCV infections only based the CAR-T cell therapy.

Author Response

We thank the reviewers for their efforts and helpful comments; we followed these suggestions and used them to further improve our manuscript.

A point by point response follows below.

Sincerely

Christopher Bednar, Armin Ensser 

Regarding the comments of Reviewer 1

This review summarizes current chimeric antigen receptor (CAR)-T cells against human cytomegalovirus (HCMV). The authors give sufficient introductions of HCMV, CARs and viral glycoproteins on the infected cells. The most interesting part is the section 4: HCMV-specific CARs, which introduces the current studies in the area. Overall, the review is written in a clear and concise manner.

Main concerns:

  1. HCMV latency and persistence. Depending on host physiology and the cell types infected, CMV persistence comprises latent, chronic, and productive states that may occur concurrently. CAR-T cell therapy of HCMV may vary during different stages, which is not well discussed.

  1. Advantages and disadvantages. The authors need to give a list or summary presenting the advantages and disadvantages on HCMV CAR-T cell therapy. Clearly, it will not eliminate HMCV infections only based the CAR-T cell therapy.

  • We have added a passage and a reference in response to both of the reviewers suggestion; we agree that in contrast to B-cell directed CAR therapy in ALL which virtually permanently eliminates the patients normal B-cells along with leukaemic cells, antiviral CARs directed at lytic viral glycoproteins surface expressed at the cell surface cannot cure latent infection, but this therapy can address lytic infection and in particular reactivation in immune compromised, hopefully until the patient is able to rise an own sufficient immune response again (lines 157-165):

“Because herpesviral protein expression is extremely downregulated in viral latency, the glycoprotein complexes are only presented on the cell surface in lytic and productive states of the infection [29]. Concluding from these observations, eradication of HCMV from infected hosts will never be achieved by CAR+ T-cells targeting the viral glycoprotein complexes due to the lack of their expression during latent stages. Nevertheless, since CAR+ T-cell therapy is supposed to be applied during severe acute lytic HCMV-infection or re-activation in immunocompromised persons, this critical lytic phase would be sup-pressed by CAR+ T-cells, thus rescuing the patient from severe sequelae or even death.”

  • In response to the reviewers suggestion, we have added a small table comparing CAR and “conventional” T cells with regard to their potential in antiviral therapy; (lines 358ff):

Reviewer 2 Report

A concise, well written review of developments of CAR T-cells targeting cytomegalovirus and other viruses. A couple of mistakes that need to be fixed are reference formatting (lines 100 and 125), a duplicate sentence in figure 1 to be removed (line 104) and typos (line 74 - enter full stop after reference; line 93 - remove full stop after 'antigen'; line 98 - replace 'T-cell' with 'T-cells')

I would like the authors to consider modifying their statement on the requirement of HLA matching for donor derived CMV specific T cells with the recipient (line 45). While HLA matching is a requirement for cytotoxic activity of ex vivo CMV specific T cells, there is now strong evidence that partial HLA matching (as low as 1 out of 10 HLA match) results in activity of ex vivo pathogen specific T cells, which allows 'cell banks' to be manufactured, cryopreserved and the same product used for many different recipients. The "labor-intensive and time-consuming effort of expanding antigen-specific T-cells" statement also applies to production of CAR T cells, the ability to make CAR T cells from any donor (and not only from the seropositive donors with a common HLA profile) is the advantage of looking into pathogen-specific CAR T cells over the ex-vivo expanded method from seropositive donors.

Author Response

We thank the reviewers for their efforts and helpful comments; we followed these suggestions and used them to further improve our manuscript.

A point by point response follows below.

Sincerely

Christopher Bednar, Armin Ensser

Regarding the comments of Reviewer 2

A concise, well written review of developments of CAR T-cells targeting cytomegalovirus and other viruses. A couple of mistakes that need to be fixed are reference formatting (lines 100 and 125), a duplicate sentence in figure 1 to be removed (line 104) and typos (line 74 - enter full stop after reference; line 93 - remove full stop after 'antigen'; line 98 - replace 'T-cell' with 'T-cells')

We have removed the duplicated sentence in the Figure 1 legend and corrected the typos (and a few more).

I would like the authors to consider modifying their statement on the requirement of HLA matching for donor derived CMV specific T cells with the recipient (line 45). While HLA matching is a requirement for cytotoxic activity of ex vivo CMV specific T cells, there is now strong evidence that partial HLA matching, (as low as 1 out of 10 HLA match) results in activity of ex vivo pathogen specific T cells, which allows 'cell banks' to be manufactured, cryopreserved and the same product used for many different recipients.
The "labor-intensive and time-consuming effort of expanding antigen-specific T-cells" statement also applies to production of CAR T cells, the ability to make CAR T cells from any donor (and not only from the seropositive donors with a common HLA profile) is the advantage of looking into pathogen-specific CAR T cells over the ex-vivo expanded method from seropositive donors.

We have added a small passage and a reference with regard to the suggestion of reviewer 2 (lines 50-58):

“However, recent studies have shown that administration of CMV-specific T-cells (CMVSTs) from a carefully selected pool of third-party donors with an HLA match of only 2 out of 8 can result in a strong antiviral response [10]. These findings allow cryopreserved ‘cell banks’ to be manufactured, at least in the case of common HLA profiles, resulting in an off-the-shelf treatment option for serious CMV-infections. Still, seropositive donors with an at least partially matching HLA-type are required to create CMVSTs, whereas T-cells carrying a CMV-specific chimeric antigen receptor (CAR), can be manufactured from any donor, including HCMV negative donors in stem cell transplantation.”

Round 2

Reviewer 1 Report

The authors fairly addressed my previous concerns.